# Comparison between the Biological Active Compounds in Plants with Adaptogenic Properties (*Rhaponticum carthamoides*, *Lepidium meyenii*, *Eleutherococcus senticosus* and *Panax ginseng*)

**DOI:** 10.3390/plants11010064

**Published:** 2021-12-26

**Authors:** Velislava Todorova, Kalin Ivanov, Stanislava Ivanova

**Affiliations:** Department of Pharmacognosy and Pharmaceutical Chemistry, Faculty of Pharmacy, Medical University-Plovdiv, 4002 Plovdiv, Bulgaria; kalin.ivanov@mu-plovdiv.bg (K.I.); stanislava.ivanova@mu-plovdiv.bg (S.I.)

**Keywords:** adaptogen, adaptogens, plant adaptogens, *Rhaponticum carthamoides*, Leuzea, ecdysterone, *Lepidium meyenii*, Maca, *Eleutherococcus senticosus*, *Panax ginseng*

## Abstract

Background: In the 1960s, research into plant adaptogens began. Plants with adaptogenic properties have rich phytochemical compositions and have been used by humanity since ancient times. However, it is not still clear whether the adaptogenic properties are because of specific compounds or because of the whole plant extracts. The aim of this review is to compare the bioactive compounds in the different parts of these plants. Methods: The search strategy was based on studies related to the isolation of bioactive compounds from *Rhaponticum carthamoides*, *Lepidium meyenii*, *Eleutherococcus senticosus*, and *Panax ginseng.* The Preferred Reporting Items for Systematic Reviews and Meta-Analyses (PRISMA) guidelines were followed. Results: This review includes data from 259 articles. The phytochemicals isolated from *Rhaponticum carthamoides*, *Lepidium meyenii*, *Eleutherococcus senticosus*, and *Panax ginseng* were described and classified in several categories. Conclusions: Plant species have always played an important role in drug discovery because their effectiveness is based on the hundreds of years of experience with folk medicine in different nations. In our view, there is great potential in the near future for some of the phytochemicals found in these plants species to become pharmaceutical agents.

## 1. Introduction

The term adaptogen was introduced for the first time in the 1940s by Dr. Nikolai Lazarev [1]. The classical definition of adaptogens is related to their ability to increase the organism’s resistance to stress factors (“stressors”) [2,3]. These stressors have different natures: chemical, physical, or other [2,3]. Examples of such stress factors include unfavorable atmospheric temperature, intense physical activity, high-altitude hypoxia, etc. According to the definition, the adaptogens should not only protect the organism from damage from stress situations/factors, but should also not increase the oxygen consumption and not disturb the normal functions of the organism [2]. Adaptogens are also called metabolic regulators [2]. However, the adaptogenic effect is described as “nonspecific” [2].

I. I. Brekhman and I. V. Dardymov are the researchers who first classified the plants with adaptogenic properties: *Panax ginseng* C. A. Mey., *Eleutherococcus senticosus* Max. and *Rhaponticum carthamoides* (Wild.) Iljin from Araliaceae family, *Rhodiola rosea* L. from Crassulaceae family and *Schisandra chinensis* from Schisandraceae family are plants with adaptogenic properties [4]. Later, the “family of plant-adaptogens” was expanded to include: *Bryonia alba* L. (Cucurbitaceae), *Tribulus terrestris* L. (Zygophyllaceae), *Bacopa monnieri* L. Pennell (Plantaginaceae), *Lepidium meyenii* Walp. (Brassicaceae) and *Withania somnifera* (L.) Dunal (Solanaceae) [3,5,6,7,8,9,10].

Plants with adaptogenic properties have rich phytochemical compositions and different applications [3,11,12,13,14,15,16,17,18,19,20]. However, it is not still clear whether the adaptogenic properties are because of specific compounds or because of the whole plant extracts. Studies that compare the biological activities of the different compounds and the whole extracts are limited. There is no study that compares the phytochemical compositions of the most important plant adaptogens.

There is great potential for some of the phytochemicals found in *Rhaponticum carthamoides*, *Lepidium meyenii*, *Eleutherococcus senticosus* and *Panax ginseng* to become pharmaceutical agents. The aim of this review is to compare the bioactive compounds in the different parts of these plants (roots, leaves, seeds, etc.), which will support the study, evaluation, and extraction of specific molecules from *Rhaponticum carthamoides*, *Lepidium meyenii*, *Eleutherococcus senticosus*, and *Panax ginseng.*

## 2. Materials and Methods

The search strategy was to seek studies related to the isolation of bioactive compounds from *Rhaponticum carthamoides*, *Lepidium meyenii*, *Eleutherococcus senticosus*, and *Panax ginseng* and their activities. This was conducted following the Preferred Reporting Items for Systematic Reviews and Meta-Analyses (PRISMA) guidelines, presented in Figure 1. This search was performed using the following databases: PubMed, Sci Finder, and Web of Science.

The keywords included in the search were: “phytosteroids”, “bioactive compounds”, “phenolic acids”, “flavonoids”, “content of ecdysterone”, “phytochemicals”, “chemical compounds”, “*Lepidium meyenii*”, “*Eleutherococcus senticosus*”, “*Rhaponticum carthamoides*”, “*Panax ginseng*”, “Maca root”, “ecdysterone”, “macamides and macaenes”, “polysaccharides”, “glucosinolates”,”eleutherosides”,”ginsenosides”, “structure-activity relationship”, “animal studies”, “human studies” and “cell culture studies*”.*

In the final step, the selected articles were read and identified. In total, 259 studies were selected and included in the present review.

## 3. Results and Discussion

The main phytochemical classes isolated from different plant parts of *Rhaponticum carthamoides*, *Lepidium meyenii*, *Eleutherococcus senticosus*, and *Panax ginseng* were phytosteroids, phytosterols, flavoloids, flavolignans, alkaloids, glucosinolates, saponins, phenolic acids and others [12,13,22,23]. Phytosteroids are the main bioactive compounds isolated from *Rhaponticum carthamoides* roots, leaves, and seeds, and they are not found in *Panax ginseng*, *Eleutherococcus senticosus*, and *Lepidium meyenii* [12,13,22,23]. Flavonoids are also detected in *Rhaponticum carthamoides* leaves and roots, but they are not detected in *Lepidium meyenii*, *Eleutherococcus senticosus* or *Panax ginseng* plant parts [12,22,23,24,25,26]. Macaenes, macamides, alkaloids, glucosinolates and sterols are isolated from *Lepidium meyenii* tuber, but are not detected in the other adaptogens, such as *Panax ginseng*, *Eleutherococcus senticosus*, and *Rhaponticum carthamoides* [12,13,22,23]. The main phytochemicals isolated from *Panax ginseng* and *Eleutherococcus senticosus* are saponins and their glycosides, which are not detected in *Rhaponticum carthamoides* and *Lepidium meyenii* [12,13,22,23]. Phenolic acids are detected in *Rhaponticum carthamoides* roots and *Eleutherococcus senticosus* roots, but not in *Lepidium meyenii* and *Panax ginseng* [22,23,27,28,29]. The isolated nutritional ingredients differentiate Maca from other plants with adaptogenic properties [12,13,22,23,30]. Maca is the only plant among these that is considered not only a medicinal plant, but also a food.

### 3.1. Rhaponticum carthamoides

*Rhaponticum carthamoides* (Wild.) Iljin is an endemic plant, naturally grown in South Siberia [13]. Humans have known of the plant since ancient times, and its various applications have been described in Eastern folk medicine [13]. It had been used for the treatment of fever, cardiovascular diseases, fatigue, kidney diseases, reproductive and sexual disfunction, quinsy, etc., [13,31,32].

*Rhaponticum carthamoides* is a perennial plant [13,31,33]. It can reach up to 150 cm in height [13]. It is a semi-rosulate plant [31]. The main parts utilized for the production of extracts are roots and rhizomes [13]. *Rhaponticum carthamoides* rhizome cum radicibus are included in Russian pharmacopoeia [34]. However, the plant is not included in the European, British, or USA pharmacopeias. The rhizome (shown in Figure 2) is dark black, vertical, branched, and wrinkled, and can reach up to 36 cm in length. The roots are smooth and elastic with numerous branches [34].

Nowadays, *Rhaponticum carthamoides* extract is included in numerous dietary supplements. Its intake is associated with not only adaptogenic activity, but also with antimicrobial, anti-oxidative, neuroprotective, antidiabetic, and anabolic activity [13,31]. The intake of *Rhapoticum carthamoides* extract is not associated with side effects [13,35,36,37,38]. The main bioactive compounds detected in *Rhaponticum carthamoides* are phytosteroids, flavonoids, and phenolic acids. The plant is also a source of essential oil [13,39,40]. The plant is a rich source of phytoecdysteroids—a large class of steroid compounds [41,42]. Their structures are composed by 27–29 C-atoms, with a four-ring steroid skeleton [39,42,43,44] and contain polyhydroxyl groups (4–7 hydroxyl groups) [45]. Nowadays, more than 200 ecdysteroid compounds are described [41,46], 50 of which are isolated from *Rhaponthicum carthamoides* [13]. These compounds are found in the roots, leaves, and seeds of the plant [47,48,49,50,51,52,53,54,55,56]. The content of 20-hydroxyecdysterone, which is the main ecdysteroid, is higher in roots than in leaves and seeds [47,48,49,50,51,52,53,54,55,56].

The structure–activity relationship of ecdisteroids is still not well clarified. Some researchers suggest that the presence of hydroxyl groups on C-5, C-14, and C-22 positions is very important for the biological activity of these compounds as well the presence of double bond at C-7 and keto-group at C-6 (example: ecdisterone) [57,58,59]. The presence of 2,3-diol system, hydroxyl group at C-20 in ecdysteroids structures is important for the anabolic activity [60].

Some of the beneficial effects related to phytoecdysteroids are anabolic, hypocholesterolemetic, neuroprotective, hypoglycemic, and metabolism regulation [60,61]. However, *Rhaponthicum carthamoides* is not the only source of ecdysteroids. There are other plants that contain 20-hydroxyecdysterone. These are *Achyranthes bidentata* Blume, *Achyranthes japonica* (Miq.) Nakai, *Ajuga iva* (L.) Schreb, Boerhaavia diffusa L. *nom. cons*., *Diploclisia glaucescens* (Blume) Diels, *Pfaffia glomerata* (Spreng.) Pedersen, *Spinacia oleracea* L., *Polypodium japonicum* Makino, and some others [62,63,64].

Since 2020, ecdysterone has been included in the World Anti-doping Agency (WADA) monitoring program [65]. According to different studies in mammals, ecdysterone has a wide variety of pharmacological effects: anabolic, anti-diabetic, anti-inflammatory, cardioprotective, hypolipidemic and others [62,66,67,68,69,70].

However, studies that investigated ecdysterone’s activity in humans are limited [36,37,38,71,72,73]. There are data about the intake of *Rhaponticum carthamoides* extract, which contains ecdysterone [38,71,73]. The intake of *Rhaponticum carthamoides* extract is associated with decreased body weight; increased resistance to disease; physical and mental endurance; improvement in cardiac and cognitive functions [35,36,72,73,74]. Studies that investigate ecdysterone activity in cell cultures are also limited [75,76,77,78,79]. According to data obtained from cell culture studies, ecdysterone has great potential to be used for the treatment of diabetes, breast cancer, Alzheimer’s disease, and osteoporosis [75,76,77,78,79].

In the near future, it is highly likely that ecdysterone will become a drug molecule, used for obesity management, reducing fatigue or the management of glucose levels. It is also highly likely to be included in WADA’s prohibited list if researchers prove it has the potential to improve athletes’ performance. However, its biological activity should be studied in more detail in cell cultures and mammals, and in randomized clinical trials.

*Rhaponthicum carthamoides* is also a source of flavonoids, which are mainly found in the roots and leaves [24,25]. Flavonoids are substances with a phenolic structure, and over 8000 flavonoids are known [80,81]. Flavonoids are divided into the subclasses flavonols, flavones, flavanones, catechins, and their glycosides [25]. The presence of flavonoids in *Rhaponticum carthamoides* extracts determines the hypolipidemic and antioxidative effects of the extract [81,82]. Antioxidant activity is associated with the presence of a large number of hydroxyl groups in flavonoids [83].

The plant is also a source of essential oil [40,84,85], which has antimicrobial, antioxidant, and anti-inflammatory activities [40,84]. Table 1 presents the bioactive compounds isolated from different plant parts of *Rhaponticum carthamoides*.

#### 3.1.1. Phytochemical Composition of *Rhaponticum carthamoides* Roots

The main compounds found in roots are phytoecdysteroids, flavonoids, phenolic acids, monoterpenes, and sesquiterpenes [26,27,40,45,47,48,49,50,51,52,55,84,86,89]. Some of the phytoecdysteroids and their derivatives isolated from the roots of *Rhaponticum carthamoides* include 20-hydroxyecdysone, leuzeasterone, polypodine B, rhapisterone, makisterone, carthamoleusterone, turkesteron, 20-hydroxyecdysone 2-acetate, 20-hydroxyecdysone 2,3, 20,22-diacetonide, 20-hydroxyecdysone 2,3-monoacetonide, 15-hydroxyponasterone A and 14-epi-ponasterone A 22-glucoside. The average concentration of 20-hydroxyecdysone (structure shown in Figure 3), also known as β-ecdysone, ecdysterone, and polypodine A, is 0.049–1.74% [51,54,80]. The flavonoids detected in *Rhaponticum carthamoides* roots are quercetin 5-*O*-galactoside and isorhamnetin 5-*O*-rhamnose [26]. Moreover, phenolic acids are detected in the roots of *Rhaponticum carthamoides* [27]. These include benzoic acid, salicylic acid, vanillic acid, chlorogenic acid, caffeic acid and gallic acid [27]. Essential oil is also isolated from *Rhaponticum carthamoides* roots [40,84,85]. Monoterpenes and sesquiterpenes are the main phytochemical elements of essential oil derived from *Rhaponticum carthamoides* [40,84]. Monoterpenes include α-pinene, β-pinene, geraniol, and limonene [40,78]. The isolated sesquiterpenes include β-caryophyllene, 13-norcypera-1(5),11(12)-diene, cyperene at a concentration of 18.2%, cadalene with a concentration of 9.6%, cyclosativene, and β-elemene [40,84]. The concentrations of extracted essential oil varied from 0.07 to 0.11% [85]. There are limited studies on the composition of *Rhaponticum carthamoides* essential oil and its therapeutic effects [3,40,84,85]. In near future, studies on essential oil and its bioactivity may increase in number.

Although the root is the main part used for the preparation of *Rhaponthicum carthamoides* extracts, other parts of the plant also have a rich phytochemical composition.

#### 3.1.2. Phytochemical Composition of *Rhaponticum carthamoides* Leaves

The main bioactive compounds discovered in the leaves of *Rhaponticum carthamoides* are phytoecdysone (20-hydroxyecdysone) and flavonoids (patuletin 3′-β-xylofuranoside and 6-hydroxykaempferol-7-*O*-(6″-*O*-acetyl-β-D-glucopyranoside)) [24,25,53,55,87,88]. The concentration of 20-hydroxyecdysone in leaves varied from 0.02 to 0.71% [53,55,87,88].

#### 3.1.3. Phytochemical Composition of *Rhaponticum carthamoides* Seeds

The phytochemicals isolated from the seeds of *Rhaponticum carthamoides* are phytosteroids and their derivates, such as 20-hydroxyecdysone, polypodine B, polypodine B-22-*O*-benzoate, makisterone A, 24(28)-dehydromakisterone A, rhapisterone, rhapisterone D 20-acetate, and carthamosterone A [53,54,56,90,91,92,93]. The average concentration of ecdysterone discovered in seeds is 0.57% [53,56,94].

### 3.2. Lepidium meyenii

*Lepidium meyenii*, known as “Maca”, is naturally grown in Peru [89]. Maca has been cultivated for more than 2000 years [18,95]. Humans have utilized the extract for the management of different conditions, such as: menopausal syndrome, impaired fertility, anemia, tuberculosis, and fatigue [95,96,97,98,99,100,101,102]. Nowadays, the plant extract is used as an aphrodisiac, anti-fatigue remedy, neuroprotector, antioxidant, memory enhancer, hormone secretion regulator, etc. [18,30,96,103,104,105]. The antiviral activity of the extract is also well-known, but its potential should be explored in more detail in the near future. The intake of Maca extract is not associated with serious side effects [101,106,107,108,109].

Maca is a perennial plant. Its overground part consists of 12–20 basal frost-hardy leaves forming a rosette, the height of which can reach 20 cm [98,110,111]. Its flowers are whitish with a length of 5 mm [98]. The fruits are two-celled [98]. The seeds are smooth and reddish with an ovoid shape [98]. The underground part—the tuber—is composed of roots and seedling stems (hypocotyl) [98,110,111,112]. The tuber color varies from white to purple; its size is usually about 10–14 cm, with different shapes [98,110,111]. The weight of the Maca tuber varies from 1 to 5 kg [111].

The bioactive compounds detected in Maca are alkaloid-like compounds, macamides, macaenes, glucosinolates, sterols, and polysaccharides [22,105,111,113]. Alkaloids are natural compounds containing basic nitrogen atoms [113]. Macamides are bioactive secondary benzylalkylamides [114]. Macaenes and macamides are polyunsaturated fatty acids and their amides [100]. They are isolated only from Maca [115,116]. The well-known bioactivities of macamides and macaenes are antitumor and antioxidant [113,117]. Phytosterols are cholesterol-like compounds isolated from plants [118]. They have a steroid structure, containing 28–29 carbon alcohols with a side chain with 9–10 carbon atoms [119]. Sterols decrease the plasma concentration of cholesterol [119]. Polysaccharides are carbohydrates. They consist of monosaccharides linked with glycoside bonds [113,120,121]. Polysaccharides are important substances with nutritional value [120,121]. The polysaccharides isolated from *Lepidium meyenii* correspond to immunomodulatory, anti-oxidant, anti-fatigue, anti-viral, anti-tussive, and anti-tumor effects [120,122]. Glucosinolates are sulfur- and nitrogen-rich organic compounds [123,124,125]. They are secondary metabolites in plants [125]. They are divided into two groups: aromatic and indolic [126,127]. Glucosinolates provide antitumor, antioxidant and fungitoxic activity [128,129,130].

Several studies involving animals have investigated the Maca extract’s biological activity, and reported some beneficial effects such as improvement of memory and cognitive functions, neuroprotective effects, regulation of sexual hormones and spermatogenesis, antioxidant activity, and improvement of lipide and glucose profiles [131,132,133,134,135,136,137]. Studies involving humans are limited. However, the data provided by these trials suggest beneficial effects of Maca extract in postmenopausal women, with the management of sexual functions and mood regulation [138,139,140,141].

The compounds isolated from Maca with the greatest potential for use as therapeutic agents are macamides and macaenes. Studies that investigate macamides’ and macaenes’ activity in humans are also limited. However, according to data obtained from animal studies, these compounds have great potential to be used for the treatment of ulcerosis, the management of exercise-induce fatigue, and the management of oxidative stress [142,143,144]. According to data obtained from cell cultures studies, macaenes and macamides have great potential to be used as antioxidants, anticancer drugs, neuroprotectors, and metabolism and inflammatory regulators [117,145].

Table 2 presents bioactive compounds and nutritional ingredients isolated from different plant parts of *Lepidium meyenii.*

#### 3.2.1. Phytochemicals Isolated from Maca Root

The main compounds isolated form this part of the plant are the macamides, imidazole alkaloids, pyrrole alkaloids, glucosinolates, flavolignans, polysaccharides, and others [30,123,127,148,151]. The main imidazole alkaloids detected in Maca root are lepidiline A and lepidiline B [148,151]. Macapyrrolins A, macapyrrolins B and macapyrrolins C are the pyrrole alkaloids detected in Maca root [123]. Glucotropaeolins, known as benzylglucosinolate and desulfoglucotropaeolin are the glucosinolates isolated from Maca root [148]. The flavolignans detected in Maca root are tricin 4′-*O* [threo-β-guaiacyl-(7″-*O*-methyl)-glyceryl] ether and tricin 4′-*O*-(erythro-β-guaiacyl-glyceryl) ether [148]. The polysaccharide MC-1 contains the following monosaccharides (with the given concentrations): arabinose—26.21%, mannose—11.81%, galactose—8.32% and glucose—53.66% [127].

#### 3.2.2. Bioactive Compounds Detected in *Lepidium meyenii* Tuber

The *Lepidium meyenii* tuber contains sterols, glucosinolates, macamides, macaenes, alkamides, and others [22,97,102,103,149]. Brassicasteryl acetate, ergosteryl acetate, campesteryl acetate, Δ22-ergostadienyl acetate, and sitosteryl acetate are sterols isolated from the *Lepidium meyenii* tuber [22]. Benzylglucosinolate and its derivate m-methoxybenzylglucosinolate are also isolated [95,143]. The alkamides discovered in tuber *Lepidium meyenii* are *N*-benzyl-9-oxo-12Z-octadecenamide, *N*-benzyl-9-oxo-12Z,15Z-octadecadienamide, *N*-benzyl-15Z-tetracosenamide, *N*-(m-methoxybenzyl) hexadecanamide and *N*-benzyl-13-oxo-9E,11E-octadecadienamide [103].

#### 3.2.3. Bioactive Compounds Isolated from *Lepidium meyenii* Hypocotyls

The hypocotyls are rich in benzylamine, benzyl glucosinolates, and their derivates, alkaloids, macamides, sterols, and phenols [104,114,115,126,146,147,150]. Some of the isolated glucosinolates and their derivatives include benzyl glucosinolate, glucoalyssin, glucosinlbin, glucobrassicin and glucobrassicanapin [150]. Some of the detected macamides are *N*-benzylhexadecaanamide, *N*-benzyloctadecanamide, *N*-benzyl-(9*Z*,12*Z*)-octadecadienamide, *N*-benzyl-(9*Z*,12*Z*,15*Z*)-octadecatrienamide, and methoxy-*N*-benzyl-(9*Z*,12*Z*,15*Z*)-octadecatrienamide [104,115,147]. The concentration of total macamides varies from 0.0016 to 0.0123% [109] Sterols isolated from Maca hypocotyls include campesterol and β-sytosterol; their structures are shown in Figure 4 [146].

#### 3.2.4. Nutritional Ingredients Isolated from Maca

*Lepidium meyenii* contains some essential nutrients, such as amino acids, fibers, fatty acids, lipids, proteins, and minerals [18,22]. Because of its unique nutritional and phytochemical composition, Maca is considered a “super food” [95,111,153]. Different Maca extracts, such as the tuber and starch, are used as food [154,155,156]. The term “super foods” includes products that are used as foods and medicine, which are edible [153]. In the last few decades, research into superfoods has increased [157]. Super foods may contain chemical-free proteins, amino acids, fatty acids, vitamins, minerals, polysaccharides, and other natural ingredients [158]. The intake of super foods provides essential nutrients and antioxidants, and it also supports the immune system, the endocrine system, and the cardiovascular system [159]. The most important nutrients from in *Lepidium meyenii* are described in Table 3.

The nutritional ingredients isolated from *Lepidium meyenii*’s roots/tuber and its hypocotyls are: proteins, oil, amino acids, fatty acids, and minerals [22,30,114]. The protein concentrations in roots and tubers varied from 10.2 to 13.42% [22,30], and the content of oil in the tuber is 1.42% [30]. The lipids concentration in the tuber is 2.2%, the concentration of hydrolysable carbohydrates is 59%, and the concentration of fibers is 8.5% [22]. The concentration of proteins in hypocotyls varied from 9.31 to 21.02%, and that in fibers varied from 17.82 to 26% [147]. The essential amino acids discovered in Maca root/tuber and hypocotyls are histidine, threonine, phenylalanine, D-phenylalanine, valin, methionine, isoleucine, leucine and lysine [22,30,155]. Non-essential amino acids isolated from Maca roots include aspartic acid, glutamic acid, serine, glycine, cysteine, alanine, arginine, tyrosine and proline [22,30,155]. Fatty acids such as lauric, C13:1 tridecanoic, myristic, palmitoleic, palmitic, linoleic, oleic, stearic, arachidic, behenic, lignoceric and nervonic are isolated from Maca root/tuber and hypocotyls [114]. The main detected minerals are Fe, Mn, Cu, Na, K, Ca, Mg, and Zn [22,30,155].

### 3.3. Eleutherococcus senticosus

*Eleutherococcus senticosus* (Araliaceae) is a small, woody shrub, known also as “Siberian ginseng”, which naturally grows in East Russia, Korea, China, and Japan [12,160]. It is a perennial plant and an important herb in Eastern folk medicine [12,161].

Nowadays, *Eleutherococcus senticosus* rhizome and radices are also considered especially valuable, and are included in the European and Russian pharmacopoeias [34,162].

The knotty *Eleutherococcus senticosus* rhizome has a diameter of 4.0 cm with an irregular cylindrical shape. The bark thickness is 2 mm with a greyish brown to blackish-brown color. The roots can be up to 15 cm in length with a diameter of 0.3 to 1.5 cm [162].

The intake of *Eleutherococcus senticosus* extract is associated with antioxidant, anti-inflammatory, adaptogenic, antidiabetic, and choleretic effects [12,160,163,164,165]. The most well-known activities of *Eleutherococus senticosus* are immunoregulation, hepatoprotection, antiviral, and antibacterial effects [12,160,163,164,165]. The intake of this extract is not associated with adverse effects [35,166,167,168].

The phytochemicals of *Eleutherococcus senticosus* roots are composed of phenylpropanoids, saponins, coumarins, lignans, polysaccharides, phenolic acids, and provitamins [164,165,169]. Saponins are natural compounds that contain an isoprenoidal-derived aglycone linked with sugar [170]. Eleutherosides provide anti-fatigue, anti-stress, anti-inflammatory, and heart-protective effects [171,172]. Coumarins are phenolic derivates with antioxidant, anti-HIV, spasmolytic, and vasodilating activity [124,125,173]. The main effect of polysaccharides is immunostimulation [169,174].

Although all parts of this plant have rich phytochemical compositions, the roots are the most utilized. Roots are used in the form of liquid extracts, powders, etc. [160,175]. According to data from human studies, *Eleutherococcus senticosus* extract has the potential to improve oxygen consumption, mental health, lipid, and glycemic profile [167,168,176,177]. Data obtained from animal studies suggests antidiabetic, antifatigue, neuroprotective, and nootropic activity [171,172,178,179].

The molecules isolated from *Eleutherococcus senticosus* with the greatest potential to become novel drug molecules are Eleutheroside B and Eleutheroside E.

Studies investigating Eleutheroside B and Eleutheroside E activity in humans are limited. However, according to data obtained from animal studies, they have great potential to be used for the treatment of inflammation, cancer, osteoporosis, and diabetes [172,180,181,182,183,184]. Studies investigating Eleutheroside B and Eleutheroside E activity in cell cultures are sparse [185]. According to data obtained from cell culture studies, they have great potential to be used for the treatment of cardiovascular diseases [185].

Table 4 shows the bioactive compounds isolated from different plant parts of *Eleutherococcus senticosus.*

#### 3.3.1. Phytochemical Compounds Isolated from *Eleutherococcus senticosus* Roots

The main detected compounds are saponins and their glycosides, polysaccharides, phenolic acids, and others [28,29,174,186,187,188,189,190,191,194]. Saponins and their glycosides isolated from *Eleutherococcus senticosus* include eleutheroside A, eleutheroside B (chemical structure shown in Figure 5) with an average concentration 0.045%, eleutheroside C, eleutheroside D, eleutheroside E with an average concentration of 0.056%, eleutheroside F and eleutheroside G [28,29,186,187,188,189,190,191]. The identified phenolic acids are chlorogenic, p-hydroxybenzoic, p-coumaric, caffeic, vanillic, and ferulic acid [28,29]. Sesamin (lignan), sytosterole (sterole) and cumarine are also isolated [28,29,194].

#### 3.3.2. Phytochemicals Isolated from *Eleutherococcus senticosus* Stem and Leaves

The main bioactive compounds are eleutheroside B, with an average concentration of 0.1203%, and eleutheroside E, with an average concentration of 0.085% [187]. Chiisanoside, hyperin, and triterpene glycosides, such as inermoside, 24-hydroxychiisanoside and 11-deoxyisochiisanoside, are the main phytochemical compounds isolated from *Eleutherococcus senticosus* leaves [192,193].

### 3.4. Panax ginseng

*Panax ginseng* has always been considered an important medicinal plant. Initially, it was an important part of Eastern folk medicine, and nowadays it is an essential pharmacopeial plant. Furthermore, in the past it was considered the most valuable of all medicinal herbs for the people of Korea, China, and Japan [195].

Brekhman was among the first researchers to introduce the novel pharmacological concept of the tonic effect of ginseng, resulting in the association of the plant with adaptogen effects [195].

Brekhman found out ginseng intake can increase non-specific resistance to various pathological or stress factors. According to his findings, the adaptogenic effect lasts for a long time, and work better under abnormal conditions (stress factors) [4,195,196].

*Panax ginseng* naturally grows in Korea and China [197,198]. The genus name “Panax” originates from Greek. The word is composed of the words “pan”, which means “all”, and “axos”, which means “treat”. The literal translation is “cure all diseases”, “cure everything” or “appropriate for treatment of every condition” [195]. The word “ginseng” has an Eastern origin [195].

It is a perennial, self-pollinating plant. It has one stalk and palmate leaves at its end. The flowering starts in its third-year growth stage. *Panax ginseng* seeds are obtained from plants no less than four years old. *Panax ginseng* roots may be white or pale yellow, and grow upright. There is one stout primary root and two or five rootlets and root hairs. The size and shape of the rootlets depends on water content, soil quality, weather, and other factors. Ginseng roots are considered most valuable between 4 and 6 years of age. Roots younger than 4 years are considered immature, and should not be used for medical purposes [195].

*Panax ginseng* radix is included in the European pharmacopoeia [199]. According to the European pharmacopeia, the root should have a cylindrical or fusiform shape with a length of 20 cm and a 2.5 cm diameter. The root surface should be pale yellow to brownish-red [199].

Nowadays, *Panax ginseng* extract is associated with antitumor, anti-fatigue, antioxidative, immunostimulating, anti-inflammation, anti-obesity, cardioprotective, antimicrobial and neuroprotective activities. The extract is also used because of its adaptogenic properties, as an antioxidant and as an aphrodisiac [195,198,200,201,202]. The intake of this extract is not associated with side effects [35,203,204,205,206].

The main active ingredients in *Panax ginseng* are saponins, also known as ginsenosides [201]. They include tetracyclic triterpenoid saponins of the dammarane type (four-ring carbon skeleton) and oleanane type (five-ring carbon skeleton) [198,201,202,207,208]. They consist of gonane, with 17 carbon atoms arranged in four rings [209]. Over 30 ginsenosides have been isolated from Panax [210]. It is considered that ginsenosides are responsible for the adaptogenic properties of *Panax ginseng* [198]. Other well-known effects of ginsenosides are related to anti-inflammatory activity, neuroprotective activity, antidiabetic effects, nootropic activity, and many other factors [207,211,212]. The variety of these activities of ginsenosides is based on the quantity and the positions of hydroxyl groups [213]. Ginsenosides can be isolated not only from *Panax ginseng*, but also from all of Panax species, such as *Panax quinquefolius* L., *Panax notoginseng* (Burkill) F. H. Chen, *Panax japonicas* (T. Nees) C. A. Mey. and *Panax zingiberensis* C. Y. Wu and K. M. Feng [207,214,215,216].

According to studies involving humans, ginsenosides may improve calmness, mental health, and the overall quality of life. Moreover, their intake is associated with antihyperlipidemic, antidiabetic, and anti-fatigue effects [203,206,217,218,219,220]. The data obtained from animal studies suggest that ginsenosides could be included in the management of diabetes and cardiovascular diseases, in the treatment of impaired immunity, or could be used as hepatoprotectors [203,221,222,223,224,225,226].

Although *Panax ginseng* is a source of plenty of biological active compounds, the molecules with the greatest potential to become drug molecules are ginsenosides. According to data obtained from animal studies, ginsenosides have great potential to be used for the treatment of cardiovascular diseases, hepatic disorders and obesity [227,228,229,230,231,232,233]. According to data obtained from cell culture studies, ginsenosides have great potential to be used for the treatment cardiovascular diseases, hypercholesterolemia, and some types of cancer [234,235,236,237].

Table 5 shows the isolated bioactive compounds from *Panax ginseng*.

#### 3.4.1. Phytochemicals Isolated from *Panax ginseng* Roots

The main phytochemicals are ginsenosides and their isomers [238,239,240,241,242,243,244,246,247,249,250,252]. Some of the ginsenosides isolated from *Panax ginseng* include ginsenoside Ra1, ginsenoside Ra2, ginsenoside Rb1, ginsenoside Rb2, ginsenoside Rb3, ginsenoside Rc, ginsenoside Rd, ginsenoside Re, ginsenoside Rh, ginseoside Rg1, ginsenoside Rg2, ginsenoside Rg5, ginsenoside Rf, ginsenoside F2, ginsenoside Rk1, ginsenoside Rs4, and ginsenoside Rs6 [238,239,240,241,242,243,244,246,247,249,250]. The average concentration of ginsenoside Ra1 in Panax ginseng roots is 0.03%, and that of ginsenoside Ra2 is 0.02% [238]. Notoginsenoside R1, notoginsenoside R2, and notoginseng R2 are also detected in *Panax ginseng* roots [239,240,245].

#### 3.4.2. Phytochemicals Isolated from *Panax ginseng* Leaves and Flower Buds

The main bioactive compounds are the ginsenosides ginsenoside Rd, ginsenoside Rh5, ginsenoside Rh6, ginsenoside Rh7, ginsenoside Rh8, ginsenoside Rh9, ginsenoside Rg1, ginsenoside Rg7 ginsenoside Re, ginsenoside F1, ginsenoside F2, and ginsenoside F3 [245,246]. The ginsenosides isolated from *Panax ginseng* may be used as melanogenic inhibitors [253]. The phytochemicals detected in flower buds are ginsenoside I and ginsenoside II [251]. The ginsenosides isolated from flower buds may be used for hepatic diseases and tumors [254,255].

## 4. Comparison between *Rhaponticum carthamoides*, *Lepidium meyenii*, *Eleutherococcus senticosus* and *Panax ginseng* and Future Perspectives

Bioactive compounds and their concentration isolated from plants are not constant. For example, the content of the phytochemicals varies in different parts of the species and also depends on many factors like soil, soil management, climate, and pollutants [55,187,256,257,258].

For that reason, it is very important the feature research about these plants to be focused mostly on their active molecules that to the whole extracts. However, comparison between the biological activity of the extracts and the active molecules would provide valuable data.

Although the four plants have quite different phytochemical composition (Table 6), the future perspectives for introduction of their specific molecules/ plant extracts as medi-cines are similar [12,13,22,23]. Most of them could be included in the management of dia-betes, cardiovascular diseases, or used as nootropic agents and hepatoprotectors (Table 7) [12,18,35,62,67,132]. *Rhaponticum carthamoides* is the only plant among these which has the greatest potential to be used as a remedy for improvement physical performance, because of potential ergogenic activity. Ecdysterone, which is one of its active compounds is in process of monitoring by WADA as a doping compound [65]. Moreover, in near future the extract or its active compounds could be applied for obesity/ overweight management [259,260].

In term to establish the biological activity of *Rhaponticum carthamoides*, *Lepidium meyenii*, *Eleutherococcus senticosus*, *Panax ginseng*/their active compounds, cell cultures research would be especially useful to give the right direction for future investigations.

## 5. Conclusions

Plants have always played an important role in drug discovery, and their effectiveness is based on hundreds of years’ experience in the folk medicines of different nations. In the 1960s, the first plants with adaptogenic activities were described: *Rhaponticum carthamoides*, *Eleutherococcus senticosus*, and *Panax ginseng.* Later, *Lepidium meyenii* was also included in the plant adaptogens family.

The main phytochemicals isolated from these plants are phytosteroids, phytosterols, alkaloids, and saponins. These biologically active compounds determine the therapeutic effects of plants not only as adaptogens, but also as antioxidants, hepatoprotectors, immunomodulators, hormone regulators, and others. Plants have always been an important source of past and novel drug molecules. In our view, there is great potential for some of the phytochemicals found in these plant species, such as ginsenosides, ecdysterone, macamides, macaenes, and eleutherosides to become novel drug molecules. However, their biological activity should be studied in more detail in cell cultures, in mammals, and in randomized clinical trials.

## Figures and Tables

**Figure 1 plants-11-00064-f001:**
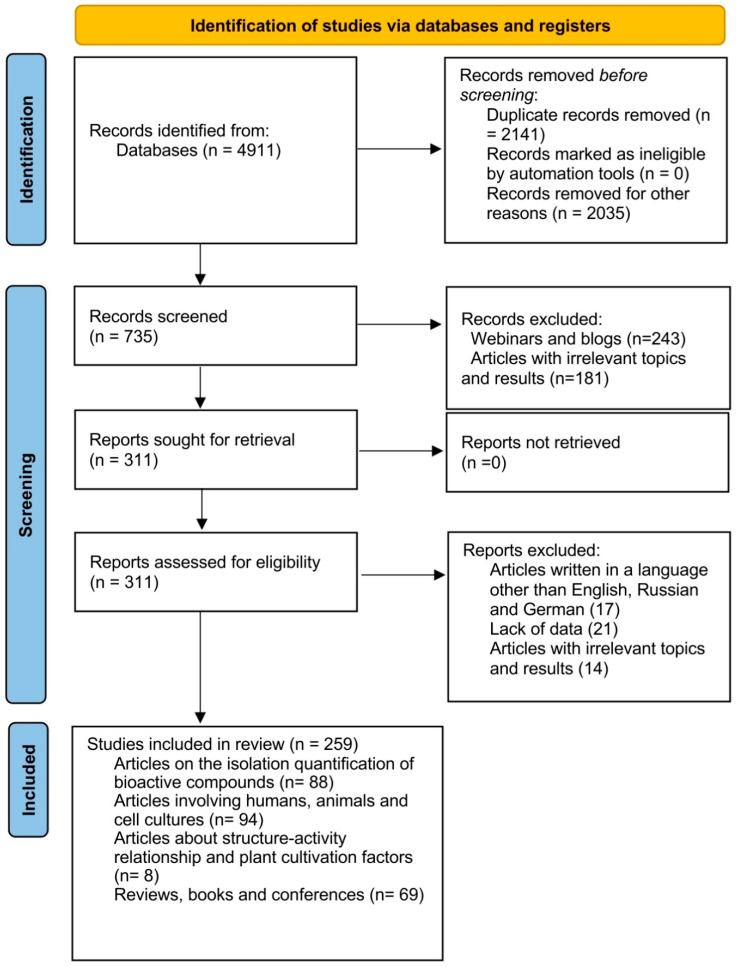
PRISMA 2020 flow diagram [21].

**Figure 2 plants-11-00064-f002:**
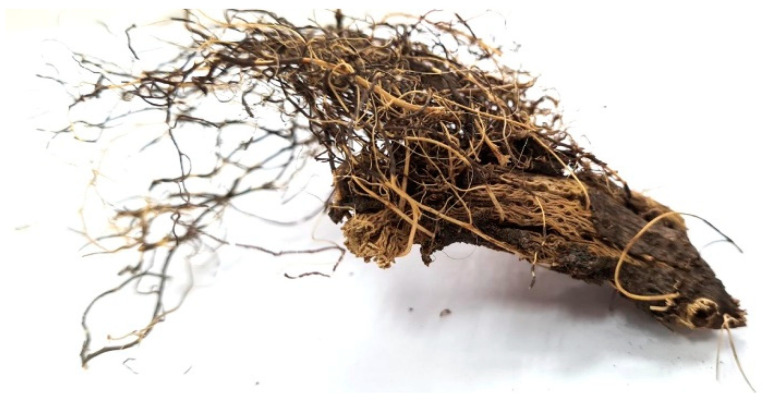
*Rhaponticum carthamoides* rhizome.

**Figure 3 plants-11-00064-f003:**
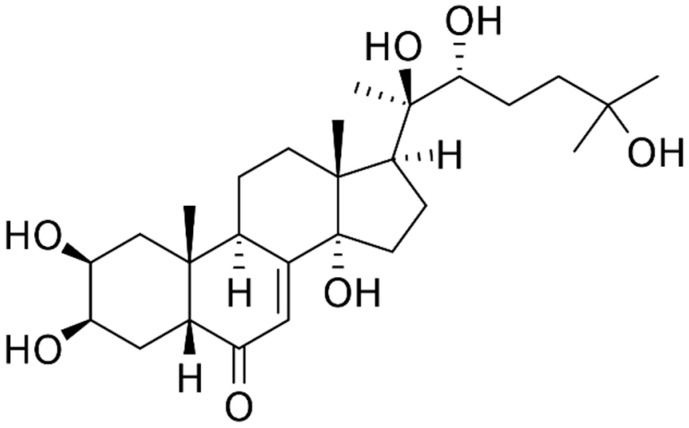
Chemical structure of 20-Hydroxyecdysone.

**Figure 4 plants-11-00064-f004:**
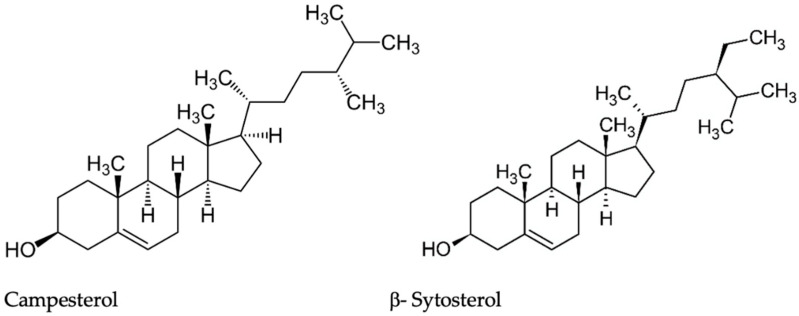
Structures of Campesterol and β-Sytosterol.

**Figure 5 plants-11-00064-f005:**
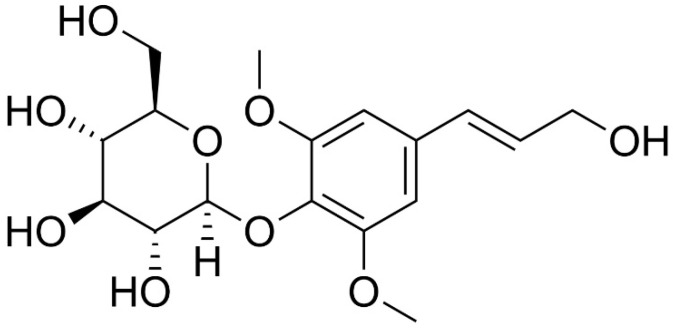
Structure of Eleutheroside B (syringine).

**Table 1 plants-11-00064-t001:** *Rhaponticum carthamoides* bioactive compounds.

Biological Active Compound	Plant Part	References
Phytosteroids		
20-Hydroxyecdysone	Roots Leaves Seeds	[47,48,49,50,51,52,55,86] [53,55,87,88] [53,54,56]
20-Hydroxyecdysone 2-acetate	Roots	[50]
20-Hydroxyecdysone 3-acetate	Roots	[50]
20-Hydroxyecdysone 2,3-monoacetonide	Roots	[48,49,86]
20-Hydroxyecdysone 20,22-monoacetonide	Roots	[48,49,86]
20-Hydroxyecdysone 2,3;20,22-diacetonide	Roots	[48,49]
2-Deoxyecdysterone	Roots	[86]
3-epi-20-Hydroxyecdysone	Roots	[49]
5-α-20-Hydroxyecdysone	Roots	[49]
22-Oxo-20-Hydroxyecdysone	Roots	[49]
Leuzeasterone	Roots	[49]
Polypodine B	Roots Seeds	[48,49,51,86] [54]
Polypodin B-22-*O*-benzoate	Seeds	[56]
Polypodine B-20,22-acetonide	Roots	[48]
Inokosterone	Roots	[50,89]
Inokosterone 20,22-acetonide	Roots	[50]
Integristerone A	Roots	[49,50,86]
Integristeone A 20,22-acetonide	Roots	[50]
Integristerone B	Roots	[49]
14-epi-Ponasterone A 22-glucoside	Roots	[50]
15-Hydroxyponasterone A	Roots	[50]
Makisterone	Roots	[51]
Makisterone A	Roots Seeds	[48,49] [90]
Makisterone C	Roots	[49,50]
24-epi-Makisterone A	Roots	[50]
24(28)-Dehydromakisterone A	Roots Seeds	[50,51,86] [54]
26-Hydroxymakisterone C	Roots	[50]
1-Hydroxymakisterone C	Roots	[50]
(24Z)-29-Hydroxy-24(28)-dehydromakisterone C	Roots	[49,50]
22-Deoxy-28-hydroxymakisterone C	Roots	[50]
Isovitexirone	Roots	[48,49]
Rhapisterone	Roots	[86]
Rhapisterone B	Seeds	[91]
Rhapisterone C	Seeds	[92]
Rhapisterone D	Seeds	[93]
Rhapisterone D 20-acetate	Seeds	[90]
Kaladasterone	Roots	[45]
5-Deoxykaladasterone	Roots	[45,51]
Munisterone A	Roots	[45]
Taxisterone	Roots	[49]
Rubrosterone	Roots	[49]
Dihydrorubrosterone	Roots	[49]
Carthamosterone	Roots	[49,50,51]
Carthamosterone A	Seeds	[54]
Ajugasterone C	Roots	[45,48,49,50,51]
Amarasterone A	Roots	[50]
24(28)-Dehydroamarasterone B	Roots	[50]
Turkesteron	Roots	[50]
Poststerone	Roots	[49]
Eriodictyol-7-β-glucopyranoside	Leaves	[24]
Flavonoids		
Quercetin 5-*O*-galactoside	Roots	[26]
Isorhamnetin 5-*O*-rhamnoside	Roots	[26]
Patuletin 3′-β-xylofuranoside	Leaves	[25]
6-Hydroxykaempferol-7-*O*-(6″-*O*-acetyl-β-D-glucopyranoside)	Leaves	[24]
Phenolic acids		
Protocatechuic acid Benzoic acid o-Hydroxyphenylacetic acid p-Hydroxyphenylacetic acid m-Hydroxybenzoic acid p-Hydroxybenzoic acid Salicylic acid Gentisic acid Elagic acid Chlorogenic acid Vanillic acid o-Coumaric acid p-Coumaric acid Synapic acid Caffeic acid Ferulic acid Gallic acid Syringic acid	Roots	[27]
Essential oil-components		
Geraniol	Roots and leaves	[85]
α-Pinene	Roots	[40,84]
β-Pinene	Roots	[84]
Limonene	Roots	[40,84]
β-Caryophyllene	Roots and leaves	[84,85]
13-Norcypera-1(5),11(12)-diene	Roots	[40]
Cyperene	Roots	[40,84]
2,5,8-Trimethyl-1-naphthol	Roots	[40]
Cadalene	Roots	[40]
Cyclosativene	Roots	[40,84]
β-Elemene	Roots	[40,84]

**Table 2 plants-11-00064-t002:** Bioactive compounds in *Lepidium meyenii*.

Biological Active Compound	Plant Part	References
Sterols		
Brassicasteryl acetate	Tuber	[22]
Ergosteryl acetate	Tuber	[22]
Campesteryl acetate	Tuber	[22]
Δ22-Ergostadienyl acetate	Tuber	[22]
Sitosteryl acetate	Tuber	[22]
Campesterol	Hypocotyls and Leaves	[146]
β-Sytosterol	Hypocotyls and Leaves	[146]
Glucosinolates		
Glucosinolate	Root	[30]
Benzyl Glucosinolate (Glucotropaeolin)	Hypocotyls Root/Tuber Fresh hypocotyls; Fresh leaf; Seed; Sprout; Dry hypocotyls	[114,126,147] [102,148,149] [150]
Desulfoglucotropaeolin	Root	[148]
m-Methoxybenzylglucosinolate	Tuber	[102,149]
5-Methylsulfinylpentyt glucosinolate (glucoalyssin)	Fresh hypocotyls; Fresh leaf; Seed; Sprout; Dry hypocotyls	[150]
p-Hydroxybenzyl glucosinolate/4-Hydroxybenzyl glucosinolate (glucosinalbin)	Fresh hypocotyls; Fresh leaf; Seed; Sprout; Dry hypocotyls	[150]
p-Hydroxybenzyl glucosinolate/4-Hydroxybenzyl glucosinolate (glucosinalbin)	Hypocotyls	[126]
m-Hydroxybenzyl-glucosinolate	Fresh hypocotyls; Fresh leaf; Seed	[150]
Pent-4-enyl glucosinolate (glucobrassicanapin)	Fresh hypocotyls; Fresh leaf	[150]
Indolyl 3-methyl glucosinolate (glucobrassicin)	Fresh hypocotyls; Fresh leaf; Dry hypocotyls	[150]
p-Methoxybenzylglucosinolate	Fresh hypocotyls; Fresh leaf; Sprout; Dry hypocotyls	[150]
4-Methoxyindolyl-3-methyl glucosinolate (4-methoxyglucobrassicin)	Fresh hypocotyls; Fresh leaf; Seed	[150]
4-Methoxyindolyl-3-methyl glucosinolate (4-methoxyglucobrassicin)	Hypocotyls	[126]
4-Hydroxy-3-indolylmethyl glucosinolate (4-Hydroxyglucobrassicin)	Hypocotyls	[126]
3-Methoxybenzyl glucosinolate (Glucolimnanthin)	Hypocotyls	[126]
5-Methylsulfinylpentyl glucosinolate (Glucoalyssin)	Hypocotyls	[126]
Alkaloids		
Total Alkaloids	Root Hypocotyls	[30] [147]
Imidazole alkaloids		
Lepidiline A (1,3-dibenzyl-4,5-dimethylimidazolium chloride)	Root	[151]
Lepidiline B (1,3-dibenzyl-2,4,5-trimethylimidazolium chloride)	Root	[148,151]
Pyrrole alkaloids		
Macapyrrolins A	Root	[123]
Macapyrrolins B	Root	[123]
Macapyrrolins C	Root	[123]
Macamides		
Macamides (benzylalkamides)	Root/Tuber	[30,97]
	Hypocotyls	[114]
Total macamides	Hypocotyls	[115]
	Hypocotyls and Leaves	[146]
*N*-benzylhexadecanamide	Hypocotyls	[104,115,147]
*N*-benzyl-(9*Z*)-octadecanamide	Hypocotyls	[104,115]
Methoxy-*N*-benzyl-(9*Z*,12*Z*)-octadecadienamide	Hypocotyls	[104]
*N*-benzyloctadecanamide	Hypocotyls	[104,115]
*N*-Benzylhexadecanamide	Hypocotyls Tuber	[115] [97]
*N*-benzyl-(9*Z*,12*Z*)-octadecadienamide	Hypocotyls	[104,115]
*N*-benzyl-(9*Z*,12*Z*,15*Z*)-octadecatrienamide	Hypocotyls	[104,115]
Methoxy-*N*-benzyl-(9*Z*,12*Z*,15*Z*)-octadecatrienamide	Hypocotyls	[104]
*N*-benzyl-5-oxo-6E,8E-octadecadienamide	Tuber	[97]
Makamide 1 (*N*-benzyl palmitamide)	Hypocotyls and Leaves	[146]
Makamide 2 (*N*-benzyl-5-oxo-6E, 8E-octadecadienamide)	Hypocotyls and Leaves	[146]
Macaridine (benzylated derivative of 1,2-dihydro-*N*-hydroxypyridine)	Tuber	[97]
Makaenes		
Makaene (5-oxo-6E,8E-octadecadienoic acid)	Tuber	[97]
Makaene (5-oxo-6E, 8E-octadecadienoic acid)	Hypocotyls and Leaves	[146]
Flavolignans		
Tricin 4′-*O* [threo-β-guaiacyl-(7″-*O*-methyl)-glyceryl] ether	Root	[148]
Tricin 4′-*O*-(erythro-β-guaiacyl-glyceryl) ether	Root	[148]
Others		
Alkamides	Tuber	[103]
Total Phenols	Hypocotyls and Leaves	[146]
Benzylamine	Hypocotyls	[114]
Tricin	Root	[148]
Pinoresinol	Root	[148]
4-Hydroxycinnamic acid	Root	[148]
Guanosine	Root	[148]
3-Hydroxybenzylisothiocyanate	Root	[148]
5-(Hydroxymethyl)-2-furfural	Root	[148]
Vanillic acid 4-*O*-β-D-glucoside	Root	[148]
Malic acid	Tuber	[102]
Malic acid benzoate	Root	[148]
Benzoyl derivative of malic acid	Tuber	[102]
Uridine acid	Tuber	[102]
Benzoyl derivates of uridine acid	Tuber	[102]
(1R,3S)-1-Methyltetrahydro-β-carboline-3-carboxylic acid	Tuber	[102]
Benzylisothiocyanate	Tuber Hypocotyls	[102] [114]
Polysaccharide MC-1	Root	[127,152]

**Table 3 plants-11-00064-t003:** Nutritional ingredients in *Lepidium meyenii*.

Nutritional Ingredient	Plant Part	References
Proteins	Root/Tuber Hypocotyls	[22,30] [147]
Oil	Root	[30]
Lipids	Tuber	[22]
Hydrolyzable carbohydrates	Tuber	[22]
Whole fibre	Tuber	[22]
Total dietary fibre	Hypocotyls	[147]
Amino acids	Root/Tuber	[22,30]
Aspartic acid	Root/Tuber	[22,30]
	Hypocotyls	[147]
Glutamic acid	Root/Tuber	[22,30]
	Hypocotyls	[147]
Serine	Root/Tuber	[22,30]
	Hypocotyls	[147]
Glycine	Root/Tuber	[22,30]
	Hypocotyls	[147]
Cysteine	Root/Tuber	[22,30]
	Hypocotyls	[147]
Alanine	Root/Tuber	[22,30]
	Hypocotyls	[147]
Arginine	Root/Tuber	[22,30]
	Hypocotyls	[147]
Tyrosine	Root/Tuber	[22,30]
	Hypocotyls	[147]
Hydroxy-Proline	Tuber	[22]
Proline	Root/Tuber	[22,30]
	Hypocotyls	[147]
Histidine	Root/Tuber	[22,30]
	Hypocotyls	[147]
Threonine	Root/Tuber	[22,30]
	Hypocotyls	[147]
Phenylalanine	Root/Tuber Hypocotyls	[22,30] [147]
D-phenylalanine	Root	[148]
Valine	Root/Tuber Hypocotyls	[22,30] [147]
Methionine	Root/Tuber Hypocotyls	[22,30] [147]
Isoleucine	Root/Tuber Hypocotyls	[22,30] [147]
Leucine	Root/Tuber Hypocotyls	[22,30] [147]
Lysine	Root/Tuber Hypocotyls	[22,30] [147]
Tryptophan	Tuber	[22]
Sarcosine	Tuber	[22]
Fatty acids	Root/Tuber	[22,114]
C12: 0 dodecanoic (lauric)	Tuber	[22]
C13:0 tridecanoic	Tuber	[22]
C13:1 7-tridecenoic	Tuber	[22]
C14:0 tetradecanoic (myristic)	Tuber	[22]
C15:0 pentadecanoic	Tuber	[22]
C15:1 7-pentadecenoic	Tuber	[22]
Cl6:0 esadecanoic (palmitic)	Tuber	[22]
C16:1 9-esadecenoic (palmitoleic)	Tuber	[22]
C17:0 heptadecanoic	Tuber	[22]
C17: l 9-heptadecenoic	Tuber	[22]
C18:0 octadecanoic (stearic)	Tuber	[22]
C18:1 9-octadecenoic (oleic)	Tuber	[22]
C18: 2 9, 12-octadecadienoic (linoleic)	Root/Tuber Hypocotyls	[22,114] [104]
C19:1 11-nonadecenoic	Tuber	[22]
Cl9:0 nonadecanoic	Tuber	[22]
C20: l 15-eicosenoic	Tuber	[22]
C20:0 eicosanoic (arachidic)	Tuber	[22]
C22:0 docosanoic (behenic)	Tuber	[22]
C24:0 tetracosanoic (lignoceric)	Tuber	[22]
C24:1 15-tetracosenoic (nervonic)	Tuber	[22]
Linolenic acid	Hypocotyls Root	[104] [114]
Minerals	Root/Tuber	[22,30]
	Hypocotyls	[147]
Fe	Root/Tuber Hypocotyls	[22,30] [147]
Mn	Root/Tuber Hypocotyls	[22,30] [147]
Cu	Root/Tuber Hypocotyls	[22,30] [147]
Na	Root/Tuber Hypocotyls	[22,30] [147]
K	Root/Tuber Hypocotyls	[22,30] [147]
Ca	Root/Tuber Hypocotyls	[22,30] [147]
Mg	Root Hypocotyls	[30] [147]
Zn	Root/Tuber Hypocotyls	[22,30] [147]

**Table 4 plants-11-00064-t004:** Bioactive compounds in *Eleutherococcus senticosus*.

Biological Active Compound	Plant Part	References
Saponins and their glycosides		
Eleutheroside A	Roots	[186]
Eleutheroside B (syringine)	Stem Roots	[187] [28,29,187,188,189]
Eleutheroside B1 (isofraxidine glucoside)	Roots	[28,29,186]
Isofraxidine—aglykone of Eleutheroside B1	Roots	[28,29]
Eleutheroside C	Roots	[186]
Eleutheroside D (syringaresinol diglucoside)	Roots	[29]
Eleutheroside E ((-)syringaresinoldiglucoside)	Stem Roots	[187] [28,187,188]
Eleutheroside E (syringaresinol di-*O*-β-D-glucoside; liriodendrin)	Roots	[189]
Eleutheroside E2	Roots	[190]
Syringaresinol (aglykone of Eleutherosde E)	Roots	[28,29]
Eleutherans A, B, C, D, E, F, G	Roots	[191]
Phenolic acids		
Chlorogenic acid	Roots	[28,29]
p-Hydroxybenzoic acid	Roots	[29]
Vanillic acid	Roots	[29]
Syringic acid	Roots	[29]
p-Coumaric acid	Roots	[29]
Caffeic acid	Roots	[29]
Ethyl ester of caffeic acid	Roots	[28]
Ferulic acid	Roots	[29]
Triterpene glycosides		
Inermoside	Leaves	[192]
1-Deoxychiisanoside	Leaves	[192]
24-Hydroxychiisanoside	Leaves	[192]
11-Deoxyisochiisanoside	Leaves	[192]
Others		
Chiisanoside	Leaves	[193]
Chiisanogenin	Leaves	[193]
Hyperin	Leaves	[193]
Isomaltol 3-*O*-alpha-D-glucopyranoside	Roots	[190]
(-) Sesamine	Roots	[28,194]
Sytoterole	Roots	[28]
Coniferine	Roots	[29]
Coniferylaldehyde	Roots	[28]
Coniferyl alcohol	Roots	[29]
Cumarine	Roots	[28]
Oleanolic acid	Roots	[28]
Polysaccharides	Roots	[174]

**Table 5 plants-11-00064-t005:** Bioactive compounds in *Panax ginseng*.

Biological Active Compound	Plant Part	References
Saponins and their glycosides		
Ginsenoside Ra1 (20(S)-protopanaxadiol 3-*O*-β-D-glucopyranosyl(1–2)-β-D-glucopyranoside-20-*O*-β-D-xylopyranosyl(1–4)-α-L-arabinosyl(1–6)-β-D-glucopyranoside)	Roots	[238,239]
Ginsenoside Ra2	Roots	[238,239]
Ginsenoside Ra3	Roots	[240,241]
Ginsenoside Rb1	Roots	[239,240,241,242,243,244]
Ginsenoside Rb2	Roots	[239,240,241,242,243,244]
Ginsenoside Rb3	Roots	[240,241,242]
Malonyl-Rb	Roots	[241]
Malonyl-Rb1	Roots	[240,241]
Ginsenodide Rc	Roots	[239,240,241,242,243,244]
Ginsenoside Rd	Leaves Roots	[245] [239,240,241,242,243,244]
Malonyl-Rd	Roots	[240]
Ginsenoside Re	Roots Leaves	[239,240,241,242,243,246] [245]
Ginsenoside Rf	Roots	[239,240,241,242,243,246,247]
20-Glc-Rf	Roots	[240]
Ginsenoside Rg1	Roots Leaves	[239,240,241,242,243,247] [245]
Ginsenoside Rg2	Roots	[239,241,246,247]
20(S)-Ginsenoside-Rg2	Roots	[240,242]
20(R)-Ginsenoside-Rg2	Roots	[242]
Ginsenoside Rg3	Roots	[239]
20(S)-Ginsenoside-Rg3	Roots	[243]
20(R)-Ginsenoside-Rg3	Roots	[242,243]
Rg3/isomer	Roots	[240]
Ginsenoside Rg5	Roots	[243]
Ginsenoside Rg6	Roots	[243]
Ginsenoside Rg7 (3-*O*-β-D-glucopyranosyl 3β,12β,20(S),24(R)-tetrahydroxy-dammar-25-ene 20-*O*-β-D-glucopyranoside)	Leaves	[248]
Ginsenoside Rh	Roots	[239,242]
Ginsenoside 20(S)-Rh1	Roots	[240,242]
Ginsenoside Rh4	Roots	[240,243]
Ginsenoside Rh5 (3β,6α,12β,24xtetrahydroxy-dammar-20(22),25-diene 6-*O*-β-D-glucopyranoside)	Leaves	[248]
Ginsnoside Rh6 (3β,6α,12β,20(S)-tetrahydroxy-25-hydroperoxy-dammar-23-ene 20-*O*-β-D-glucopyranoside)	Leaves	[248]
Ginsenoside Rh7 (3β,7β,12β,20(S)-tetrahydroxy-dammar-5,24-diene 20-*O*-β-D-glucopyranoside)	Leaves	[248]
Ginsenoside Rh8 (3β,6α,20(S)-trihydroxy-dammar-24-ene-12-one 20-*O*-β-D-glucopyranoside)	Leaves	[248]
Ginsenoside Rh9 (3β,6α,20(S)-trihydroxy-12b,23-epoxy-dammar-24-ene 20-*O*-β-D-glucopyranoside)	Leaves	[248]
Ginsenoside Rk1	Roots	[243,249]
Ginsenoside Rk2	Roots	[249]
Ginsenoside Rk3	Roots	[243,249]
Ginsenoside Ro	Roots	[239,241,242,244]
Ginsenoside Ro isomer	Roots	[240]
Polyacetyleneginsenoside-Ro	Roots	[247]
Ginsenoside-Ro methyl ester	Roots	[247]
Ginsenoside Rs1	Roots	[242]
20(S)-Ginsenoside Rs3	Roots	[243]
20(R)-Ginsenoside Rs3	Roots	[243]
Ginsenoside Rs4 (3β,12β-dihydroxydammar-20(22),24-diene-3-*O*-β-D-glucopyranosyl(1→2)-P-D-6″-*O*-acetylglucopyranoside)	Roots	[243,250]
Ginsenoside Rs5 (3β,12β-dihydroxydammar-20(21), 24-diene-3-*O*-β-D-glucopyranosyl(1→2)-β-D-6″-*O*-acetylglucopyranoside)	Roots	[243,250]
Ginsenoside Rs6 (3β, 6α,12p-trihydro-xydammar-20(22),24-diene-6-*O*-β-D-6′-*O*-acetylglucopyranoside)	Roots	[250]
Ginsenoside Rs7 (3β,6α, 12β-trihydroxydam-mar-20(21),24-diene-6-*O*-β-D-6′-*O*-acetylglucopyranoside)	Roots	[250]
Ginsenoside F1 (20-*O*-β-glucopyranosyl-20(S)-protopanaxatriol)	Leaves	[245]
Ginsenoside F2 (3, 20-di-*O*-β-glucopyranosyl-20(S)-protopanaxadiol)	Leaves	[245]
Ginsenoside F3 (20-*O*-(α-arabinopyranosyl-(1→6)-β-glucopyranosyl)-20(S)-protopanaxatriol)	Leaves	[245]
Ginsenoside I	Flower buds	[251]
Ginsenoside II	Flower buds	[251]
Ginsenoside F4	Roots	[243]
Malonyl-Ra1/Ra2	Roots	[241]
Malonyl-Rb2/Rb3/Rc	Roots	[241]
Malonyl-Rd Notoginsenoside R2/F3	Roots	[241]
Malonyl-Rd isomer	Roots	[241]
Ra1/Ra2/isomer	Roots	[240,241]
Gingerglycolipid B	Roots	[247]
Quinginsenoside R1	Roots	[242]
Koryoginsenoside-R1 (6-*O*-[trans butenoyl-(1→6)-β-D-glucopyranosyl]-20-*O*-β-D-glucopyranosyl dammar-24-en-3β, 6α,12β,20(S)-tetrol)	Roots	[239]
Koryoginsenoside-R2 3-*O*-[β-D-glucopyranosyl-(1→2)-β-D-glucopyranosyl]-20-*O*-[β-D-glucopyranosyl-(1→6)-β-D-glucopyranosyl] dammar-22-en-3β, 12β, 20(S), -25-tetrol	Roots	[239]
Notoginsenoside R1	Roots	[239]
Notoginsenoside R2	Roots	[240]
Notoginseng R2	Roots	[252]
Malonyl-Rg1	Roots	[240]
Malonyl-Rc/Rb2/Rb3	Roots	[240]
Rg6/F4	Roots	[240]
Rg5/Rk1	Roots	[240]

**Table 6 plants-11-00064-t006:** Comparison between the main bioactive compounds in *Rhaponticum carthamoides*, *Lepidium meyenii*, *Eleutherococcus senticosus*, and *Panax ginseng*.

Bioactive Compounds	*Rhaponticum* *carthamoides*	*Lepidium meyenii*	*Eleutherococcus senticosus*	*Panax ginseng*
Phytosteroids	[24,47,48,49,50,51,52,53,54,55,56,86,87]	-	-	-
Glucosinolates	-	[30,102,114,126,147,148,149,150]	-	-
Alkaloids	-	[30,123,147,148,151]	-	-
Macamides and makaaenes	-	[30,97,104,114,115,146,147,148]	-	-
Eleutherosides	-	[127,152]	[28,29,186,187,188,190,191,192]	-
Ginsenosides	-	-	-	[238,239,240,241,242,243,244,245,246,247,248,249,250,251,252]

**Table 7 plants-11-00064-t007:** Effects and future perspectives of extracts/ bioactive compounds.

Effects/Activity	*Rhaponthicum* *carthamoides*	*Lepidium meyenii*	*Eleutherococcus* *senticosus*	*Panax ginseng*
Weight loss management	+	-	-	-
Lipid profile management	+	+	+	+
Nootropic activity	+	+	+	+
Diabetes management	+	+	+	+
Ergogenic activity	More data are needed. In process of monitoring	-	-	-
Hormones regulation	+	+	-	-
Antiviral activity	More data are needed	More data are needed	More data are needed	More data are needed

## Data Availability

Not applicable.

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
