# Peer review of "Comparison between the Biological Active Compounds in Plants with Adaptogenic Properties (Rhaponticum carthamoides, Lepidium meyenii, Eleutherococcus senticosus and Panax ginseng)"

_plants, 2021, doi:10.3390/plants11010064_

Round 1
Reviewer 1 Report
There are several benefits from the immense natural chemical diversity and several successes have been recorded in finding new actives in natural products, some of which have become new drugs. And in this review, the authors present several examples of plants containing bioactive phytochemicals. The authors list their bioactivity isolation and their importance in drug discovery. Henceforth, a comprehensive report on this topic would be of great benefit to the scientific community in general. It is recommended for publication in Plants.
Author Response
Reviewer 1 comments:
There are several benefits from the immense natural chemical diversity and several successes have been recorded in finding new actives in natural products, some of which have become new drugs. And in this review, the authors present several examples of plants containing bioactive phytochemicals. The authors list their bioactivity isolation and their importance in drug discovery. Henceforth, a comprehensive report on this topic would be of great benefit to the scientific community in general. It is recommended for publication in Plants.
Authors’ response:
Dear reviewer,
Thank you for your review.
Reviewer 2 comments:
After reading the review Comparison between the biological active compounds in plants with adaptogenic properties (Rhaponticum carthamoides, Lepidium meyenii, Eleutherococcus senticosus and Panax ginseng) I found the issues set out in the writing appropriate and of importance, the only thing I suggest is to attach a perspective section where they detail based on their research that you can continue to investigate these plants.
Authors’ response:
Dear reviewer,
Thank you for your recommendations. We have added new section 4. Comparison between Rhaponticum carthamoides, Lepidium meyenii, Eleutherococcus senticosus and Panax ginseng and future perspectives.
Bioactive compounds and their concentration isolated from plants are not constant. For example, the content of the phytochemicals varies in different parts of the species and also depends on many factors like soil, soil management, climate and pollutants [55,187,256–258].
For that reason, it is very important the feature researches about these plants to be focused mostly on their active molecules that to the whole extracts. However, comparison between the biological activity of the extracts and the active molecules would provide valuable data.
Although the four plants have quite different phytochemical composition (Table 6) the future perspectives for introduction of their specific molecules/ plant extracts as medi-cines are similar [12,13,22,23]. Most of them could be included in the management of dia-betes, cardiovascular diseases or used as nootropic agents and hepatoprotectors (Table 7) [12,18,35,62,67,132]. Rhaponticum carthamoides is the only plant among these which has the greatest potential to be used as a remedy for improvement physical performance, be-cause of potential ergogenic activity. Ecdysterone, which is one of its active compounds is in process of monitoring by WADA as a doping compound [65]. Moreover, in near future the extract or its active compounds could be applied for obesity/ overweight management.
Table 7. Effects and future perspectives of extracts/ bioactive compounds
|
Effects/ activity |
Rhaponthicum carthamoides |
Lepidium meyenii |
Eleutherococcus senticosus |
Panax ginseng |
|
Weight loss management |
+ |
- |
- |
- |
|
Lipid profile management |
+ |
+ |
+ |
+ |
|
Nootropic activity |
+ |
+ |
+ |
+ |
|
Diabetes management |
+ |
+ |
+ |
+ |
|
Ergogenic activity |
More data is needed. In process of monitoring |
- |
- |
- |
|
Hormones regulation |
+ |
+ |
- |
- |
|
Antiviral activity |
More data is needed |
More data is needed |
More data is needed |
More data is needed |
In term to establish the biological activity of Rhaponticum carthamoides, Lepidium meyenii, Eleutherococcus senticosus, Panax ginseng/their active compounds, cell cultures researches would be especially useful to give the right direction for future investigations.
Reviewer 3 comments:
- Reviewer 3 comment 1:
The authors wished to make a Comparison between the biological active compounds in plants with adaptogenic properties (Rhaponticum carthamoides, Lepidium meyenii, Eleutherococcus senticosus and Panax ginseng). Please see below my suggestions regarding your manuscript.
First of all, I want to mention that in a Review type study, the titles of the main sections can be reshaped accordingly, not necessary maintaining the main titles as Introduction, Material and methods, Results and Conclusions - as for an original Article.
- Authors’ response:
Dear reviewer,
Thank you for this recommendation. We would like our review to include the different sections because it is friendly for reading. This would help reader to reach easier the information he needs.
- Reviewer 3 comment 2:
L55-58. Please highlight the novelty that this paper brings to the field, as to draw the attention of those interested on this Review, or underline the special aspects that have been addressed through the manuscript. In the actual shape, I cannot see the relevance of your Review from this aim of the study
- Authors’ response:
Dear reviewer,
Thank you for this remark. This is the first manuscript which directly compared the phytochemical composition of Rhaponticum carthamoides, Lepidium meyenii, Eleutherococcus senticosus and Panax ginseng and it would support the future investing about adaptogens with plant origin.
- Reviewer 3 comment 3:
L71-80 Information provided (inclusion/exclusion criteria) here must be mentioned in the PRISMA flow chart, removing the duplicate information (i.e. L77, 79, etc.). This is exactly the purpose of the PRISMA diagram: to summarise the description of references selection in a single schematised figure. Detail better the PRISMA figure.
- Authors’ response:
Dear reviewer,
Thank you for this remark. We have removed the duplicated information that you have mentioned. We included inclusion and exclusion criteria in PRISMA flow chart- Figure 1.
- Reviewer 3 comment 4:
L102. It must be completed that the active compounds identified in the plants you have focused have specific actions, roles and proprieties, strongly correlated with their chemical structure, as it was well stated by Glevitzky I., et al. Statistical Analysis of the Relationship Between Antioxidant Activity and the Structure of Flavonoid Compounds. Rev. Chim. 2019, 70(9), 3103-3107. https://doi.org/10.37358/RC.19.9.7497
Furthermore, the content in active compounds are also strong related to the conditions for their cultivation/growing (climate, type/characteristics of the soil/soil management, harvesting time, etc). I suggest checking and referring to Samuel A.D. et al.., Enzymological and physicochemical evaluation of the effects of soil management practices, Rev. Chim. 2017, 68(10) 2243-2247. https://doi.org/10.37358/RC.17.10.5864 ; Bungau et al. Expatiating the impact of anthropogenic aspects and climatic factors on long term soil monitoring and management. Environ Sci. Pollut. Res. 2021, 202, 30528-30550. https://doi.org/10.1007/s11356-021-14127-7
- Authors’ response:
Dear reviewer,
Thank you for this remark. We have included structure-activity relationship and relation between compounds and cultivation.
The structure-activity relationship of ecdisteroids is still not well clarified. Some researchers suggest that the presence of hydroxyl groups on C-5, C-14 and C-22 positions is very important for the biological activity of these compounds as well the presence of double bond at C- 7 and keto-group at C-6 (example: ecdisterone) [57–59]. The presence of 2,3-diol system, hydroxyl group at C-20 in ecdysteroids structures is important for the anabolic ac-tivity [60].
Antioxidant activity is associated with the presence of a large number of hydroxyl groups in flavonoids [83].
The variety of these activities of ginsenosides is based on the quantity and the positions of hydroxyl groups [213].
Bioactive compounds and their concentration isolated from plants are not constant. For example, the content of the phytochemicals vary in different parts of the species and also depends on many factors like soil, soil management, climate and pollutants [55,187,256–258].
For that reason, it is very important the feature researches about these plants to be focused mostly on their active molecules that to the whole extracts. However, comparison between the biological activity of the extracts and the active molecules would provide valuable data.
The recommended references are included in the manuscript:
- Glevitzky, I.; Dumitrel, G.A.; Glevitzky, M.; Pasca, B.; Otrisal, P.; Bungau, S.; Cioca, G.; Pantis, C.; Popa, M. Statistical Analysis of the Relationship Between Antioxidant Activity and the Structure of Flavonoid Compounds. Revista de Chimie 2019, 70, 3103–3107.
- Samuel, A.D.; Tit, D.M.; Melinte (Frunzulica), C.E.; Iovan, C.; Purza, L.; Gitea, M.; Bungau, S. Enzymological and Physicochemical Evaluation of the Effects of Soil Management Practices. Revista de Chimie 2017, 68, 2243–2247.
- Bungau, S.; Behl, T.; Aleya, L.; Bourgeade, P.; Aloui-Sossé, B.; Purza, A.L.; Abid, A.; Samuel, A.D. Expatiating the Impact of Anthropogenic Aspects and Climatic Factors on Long-Term Soil Monitoring and Management. Environ Sci Pollut Res 2021, 28, 30528–30550, doi:10.1007/s11356-021-14127-7.
- Reviewer 3 comment 5:
Before Conclusion, I consider you must add a new section related exactly to the title: Comparison of the four plants, which is mandatory, as in the actual shape of the manuscript is no comparison, just data about each plant, separately. A summarising Figure related to the effects/actions of all 4 plants and a table making comparison between their main chemical active compounds would be very helpful, increasing the relevance of the text.
- Authors’ response:
Dear reviewer,
Thank you for this remark. We have added new section: 4. Comparison between Rhaponticum carthamoides, Lepidium meyenii, Eleutherococcus senti-cosus and Panax ginseng and future perspectives.
We added table summarizing effects and future perspectives of extracts/ bioactive compounds of the four plants and table- Comparison between the main bioactive compounds in Rhaponticum carthamoides, Lepidium meyenii, Eleutherococcus senticosus and Panax ginseng.
Although the four plants have quite different phytochemical composition (Table 6) the fu-ture perspectives for introduction of their specific molecules/ plant extracts as medicines are similar [12,13,22,23]. Most of them could be included in the management of diabetes, cardiovascular diseases or used as nootropic agents and hepatoprotectors (Table 7) [12,18,35,62,67,132]. Rhaponticum carthamoides is the only plant among these which has the greatest potential to be used as a remedy for improvement physical performance, be-cause of potential ergogenic activity. Ecdysterone, which is one of its active compounds is in process of monitoring by WADA as a doping compound [65]. Moreover, in near future the extract or its active compounds could be applied for obesity/ overweight management.
Table 6. Comparison between the main bioactive compounds in Rhaponticum carthamoides, Lepidium meyenii, Eleutherococcus senticosus and Panax ginseng
|
Bioactive compounds |
Rhaponticum carthamoides |
Lepidium meyenii |
Eleutherococcus senticosus |
Panax ginseng |
|||
|
Phytosteroids |
[24,47–56,86,87] |
- |
- |
- |
|||
|
Flavonoids |
[24–26] |
- |
- |
- |
|||
|
Sterols |
- |
[22,146] |
- |
- |
|||
|
Glucosinolates |
- |
[30,102,114,126,147–150] |
- |
- |
|||
|
Alkaloids |
- |
[30,123,147,148,151] |
- |
- |
|||
|
Macamides and makaaenes |
- |
[30,97,104,114,115,146–148] |
- |
- |
|||
|
Eleutherosides |
- |
[127,152] |
[28,29,186–188,190–192] |
- |
|||
|
Ginsenosides |
- |
- |
- |
[238–241,243–252,261] |
|||
|
Phenolic acids |
[27] |
- |
[28,29] |
- |
|||
|
Polysaccharides |
- |
- |
[174] |
- |
|||
Table 7. Effects and future perspectives of extracts/ bioactive compounds
|
Effects/ activity |
Rhaponthicum carthamoides |
Lepidium meyenii |
Eleutherococcus senticosus |
Panax ginseng |
|
Weight loss management |
+ |
- |
- |
- |
|
Lipid profile management |
+ |
+ |
+ |
+ |
|
Nootropic activity |
+ |
+ |
+ |
+ |
|
Diabetes management |
+ |
+ |
+ |
+ |
|
Ergogenic activity |
More data is needed. In process of monitoring |
- |
- |
- |
|
Hormones regulation |
+ |
+ |
- |
- |
|
Antiviral activity |
More data is needed |
More data is needed |
More data is needed |
More data is needed |
In term to establish the biological activity of Rhaponticum carthamoides, Lepidium meyenii, Eleutherococcus senticosus, Panax ginseng/their active compounds, cell cultures researches would be especially useful to give the right direction for future investigations.

Reviewer 2 Report
After reading the review Comparison between the biological active compounds in plants with adaptogenic properties (Rhaponticum carthamoides, Lepidium meyenii, Eleutherococcus senticosus and Panax ginseng) I found the issues set out in the writing appropriate and of importance, the only thing I suggest is to attach a perspective section where they detail based on their research that you can continue to investigate these plants.Author Response
Reviewer 1 comments:
There are several benefits from the immense natural chemical diversity and several successes have been recorded in finding new actives in natural products, some of which have become new drugs. And in this review, the authors present several examples of plants containing bioactive phytochemicals. The authors list their bioactivity isolation and their importance in drug discovery. Henceforth, a comprehensive report on this topic would be of great benefit to the scientific community in general. It is recommended for publication in Plants.
Authors’ response:
Dear reviewer,
Thank you for your review.
Reviewer 2 comments:
After reading the review Comparison between the biological active compounds in plants with adaptogenic properties (Rhaponticum carthamoides, Lepidium meyenii, Eleutherococcus senticosus and Panax ginseng) I found the issues set out in the writing appropriate and of importance, the only thing I suggest is to attach a perspective section where they detail based on their research that you can continue to investigate these plants.
Authors’ response:
Dear reviewer,
Thank you for your recommendations. We have added new section 4. Comparison between Rhaponticum carthamoides, Lepidium meyenii, Eleutherococcus senticosus and Panax ginseng and future perspectives.
Bioactive compounds and their concentration isolated from plants are not constant. For example, the content of the phytochemicals varies in different parts of the species and also depends on many factors like soil, soil management, climate and pollutants [55,187,256–258].
For that reason, it is very important the feature researches about these plants to be focused mostly on their active molecules that to the whole extracts. However, comparison between the biological activity of the extracts and the active molecules would provide valuable data.
Although the four plants have quite different phytochemical composition (Table 6) the future perspectives for introduction of their specific molecules/ plant extracts as medi-cines are similar [12,13,22,23]. Most of them could be included in the management of dia-betes, cardiovascular diseases or used as nootropic agents and hepatoprotectors (Table 7) [12,18,35,62,67,132]. Rhaponticum carthamoides is the only plant among these which has the greatest potential to be used as a remedy for improvement physical performance, be-cause of potential ergogenic activity. Ecdysterone, which is one of its active compounds is in process of monitoring by WADA as a doping compound [65]. Moreover, in near future the extract or its active compounds could be applied for obesity/ overweight management.
Table 7. Effects and future perspectives of extracts/ bioactive compounds
|
Effects/ activity |
Rhaponthicum carthamoides |
Lepidium meyenii |
Eleutherococcus senticosus |
Panax ginseng |
|
Weight loss management |
+ |
- |
- |
- |
|
Lipid profile management |
+ |
+ |
+ |
+ |
|
Nootropic activity |
+ |
+ |
+ |
+ |
|
Diabetes management |
+ |
+ |
+ |
+ |
|
Ergogenic activity |
More data is needed. In process of monitoring |
- |
- |
- |
|
Hormones regulation |
+ |
+ |
- |
- |
|
Antiviral activity |
More data is needed |
More data is needed |
More data is needed |
More data is needed |
In term to establish the biological activity of Rhaponticum carthamoides, Lepidium meyenii, Eleutherococcus senticosus, Panax ginseng/their active compounds, cell cultures researches would be especially useful to give the right direction for future investigations.
Reviewer 3 comments:
- Reviewer 3 comment 1:
The authors wished to make a Comparison between the biological active compounds in plants with adaptogenic properties (Rhaponticum carthamoides, Lepidium meyenii, Eleutherococcus senticosus and Panax ginseng). Please see below my suggestions regarding your manuscript.
First of all, I want to mention that in a Review type study, the titles of the main sections can be reshaped accordingly, not necessary maintaining the main titles as Introduction, Material and methods, Results and Conclusions - as for an original Article.
- Authors’ response:
Dear reviewer,
Thank you for this recommendation. We would like our review to include the different sections because it is friendly for reading. This would help reader to reach easier the information he needs.
- Reviewer 3 comment 2:
L55-58. Please highlight the novelty that this paper brings to the field, as to draw the attention of those interested on this Review, or underline the special aspects that have been addressed through the manuscript. In the actual shape, I cannot see the relevance of your Review from this aim of the study
- Authors’ response:
Dear reviewer,
Thank you for this remark. This is the first manuscript which directly compared the phytochemical composition of Rhaponticum carthamoides, Lepidium meyenii, Eleutherococcus senticosus and Panax ginseng and it would support the future investing about adaptogens with plant origin.
- Reviewer 3 comment 3:
L71-80 Information provided (inclusion/exclusion criteria) here must be mentioned in the PRISMA flow chart, removing the duplicate information (i.e. L77, 79, etc.). This is exactly the purpose of the PRISMA diagram: to summarise the description of references selection in a single schematised figure. Detail better the PRISMA figure.
- Authors’ response:
Dear reviewer,
Thank you for this remark. We have removed the duplicated information that you have mentioned. We included inclusion and exclusion criteria in PRISMA flow chart- Figure 1.
- Reviewer 3 comment 4:
L102. It must be completed that the active compounds identified in the plants you have focused have specific actions, roles and proprieties, strongly correlated with their chemical structure, as it was well stated by Glevitzky I., et al. Statistical Analysis of the Relationship Between Antioxidant Activity and the Structure of Flavonoid Compounds. Rev. Chim. 2019, 70(9), 3103-3107. https://doi.org/10.37358/RC.19.9.7497
Furthermore, the content in active compounds are also strong related to the conditions for their cultivation/growing (climate, type/characteristics of the soil/soil management, harvesting time, etc). I suggest checking and referring to Samuel A.D. et al.., Enzymological and physicochemical evaluation of the effects of soil management practices, Rev. Chim. 2017, 68(10) 2243-2247. https://doi.org/10.37358/RC.17.10.5864 ; Bungau et al. Expatiating the impact of anthropogenic aspects and climatic factors on long term soil monitoring and management. Environ Sci. Pollut. Res. 2021, 202, 30528-30550. https://doi.org/10.1007/s11356-021-14127-7
- Authors’ response:
Dear reviewer,
Thank you for this remark. We have included structure-activity relationship and relation between compounds and cultivation.
The structure-activity relationship of ecdisteroids is still not well clarified. Some researchers suggest that the presence of hydroxyl groups on C-5, C-14 and C-22 positions is very important for the biological activity of these compounds as well the presence of double bond at C- 7 and keto-group at C-6 (example: ecdisterone) [57–59]. The presence of 2,3-diol system, hydroxyl group at C-20 in ecdysteroids structures is important for the anabolic ac-tivity [60].
Antioxidant activity is associated with the presence of a large number of hydroxyl groups in flavonoids [83].
The variety of these activities of ginsenosides is based on the quantity and the positions of hydroxyl groups [213].
Bioactive compounds and their concentration isolated from plants are not constant. For example, the content of the phytochemicals vary in different parts of the species and also depends on many factors like soil, soil management, climate and pollutants [55,187,256–258].
For that reason, it is very important the feature researches about these plants to be focused mostly on their active molecules that to the whole extracts. However, comparison between the biological activity of the extracts and the active molecules would provide valuable data.
The recommended references are included in the manuscript:
- Glevitzky, I.; Dumitrel, G.A.; Glevitzky, M.; Pasca, B.; Otrisal, P.; Bungau, S.; Cioca, G.; Pantis, C.; Popa, M. Statistical Analysis of the Relationship Between Antioxidant Activity and the Structure of Flavonoid Compounds. Revista de Chimie 2019, 70, 3103–3107.
- Samuel, A.D.; Tit, D.M.; Melinte (Frunzulica), C.E.; Iovan, C.; Purza, L.; Gitea, M.; Bungau, S. Enzymological and Physicochemical Evaluation of the Effects of Soil Management Practices. Revista de Chimie 2017, 68, 2243–2247.
- Bungau, S.; Behl, T.; Aleya, L.; Bourgeade, P.; Aloui-Sossé, B.; Purza, A.L.; Abid, A.; Samuel, A.D. Expatiating the Impact of Anthropogenic Aspects and Climatic Factors on Long-Term Soil Monitoring and Management. Environ Sci Pollut Res 2021, 28, 30528–30550, doi:10.1007/s11356-021-14127-7.
- Reviewer 3 comment 5:
Before Conclusion, I consider you must add a new section related exactly to the title: Comparison of the four plants, which is mandatory, as in the actual shape of the manuscript is no comparison, just data about each plant, separately. A summarising Figure related to the effects/actions of all 4 plants and a table making comparison between their main chemical active compounds would be very helpful, increasing the relevance of the text.
- Authors’ response:
Dear reviewer,
Thank you for this remark. We have added new section: 4. Comparison between Rhaponticum carthamoides, Lepidium meyenii, Eleutherococcus senti-cosus and Panax ginseng and future perspectives.
We added table summarizing effects and future perspectives of extracts/ bioactive compounds of the four plants and table- Comparison between the main bioactive compounds in Rhaponticum carthamoides, Lepidium meyenii, Eleutherococcus senticosus and Panax ginseng.
Although the four plants have quite different phytochemical composition (Table 6) the fu-ture perspectives for introduction of their specific molecules/ plant extracts as medicines are similar [12,13,22,23]. Most of them could be included in the management of diabetes, cardiovascular diseases or used as nootropic agents and hepatoprotectors (Table 7) [12,18,35,62,67,132]. Rhaponticum carthamoides is the only plant among these which has the greatest potential to be used as a remedy for improvement physical performance, be-cause of potential ergogenic activity. Ecdysterone, which is one of its active compounds is in process of monitoring by WADA as a doping compound [65]. Moreover, in near future the extract or its active compounds could be applied for obesity/ overweight management.
Table 6. Comparison between the main bioactive compounds in Rhaponticum carthamoides, Lepidium meyenii, Eleutherococcus senticosus and Panax ginseng
|
Bioactive compounds |
Rhaponticum carthamoides |
Lepidium meyenii |
Eleutherococcus senticosus |
Panax ginseng |
|||
|
Phytosteroids |
[24,47–56,86,87] |
- |
- |
- |
|||
|
Flavonoids |
[24–26] |
- |
- |
- |
|||
|
Sterols |
- |
[22,146] |
- |
- |
|||
|
Glucosinolates |
- |
[30,102,114,126,147–150] |
- |
- |
|||
|
Alkaloids |
- |
[30,123,147,148,151] |
- |
- |
|||
|
Macamides and makaaenes |
- |
[30,97,104,114,115,146–148] |
- |
- |
|||
|
Eleutherosides |
- |
[127,152] |
[28,29,186–188,190–192] |
- |
|||
|
Ginsenosides |
- |
- |
- |
[238–241,243–252,261] |
|||
|
Phenolic acids |
[27] |
- |
[28,29] |
- |
|||
|
Polysaccharides |
- |
- |
[174] |
- |
|||
Table 7. Effects and future perspectives of extracts/ bioactive compounds
|
Effects/ activity |
Rhaponthicum carthamoides |
Lepidium meyenii |
Eleutherococcus senticosus |
Panax ginseng |
|
Weight loss management |
+ |
- |
- |
- |
|
Lipid profile management |
+ |
+ |
+ |
+ |
|
Nootropic activity |
+ |
+ |
+ |
+ |
|
Diabetes management |
+ |
+ |
+ |
+ |
|
Ergogenic activity |
More data is needed. In process of monitoring |
- |
- |
- |
|
Hormones regulation |
+ |
+ |
- |
- |
|
Antiviral activity |
More data is needed |
More data is needed |
More data is needed |
More data is needed |
In term to establish the biological activity of Rhaponticum carthamoides, Lepidium meyenii, Eleutherococcus senticosus, Panax ginseng/their active compounds, cell cultures researches would be especially useful to give the right direction for future investigations.

Reviewer 3 Report
The authors wished to make a Comparison between the biological active compounds in plants with adaptogenic properties (Rhaponticum carthamoides, Lepidium meyenii, Eleutherococcus senticosus and Panax ginseng). Please see below my suggestions regarding your manuscript.
First of all, I want to mention that in a Review type study, the titles of the main sections can be reshaped accordingly, not necessary maintaining the main titles as Introduction, Material and methods, Results and Conclusions - as for an original Article.
L55-58. Please highlight the novelty that this paper brings to the field, as to draw the attention of those interested on this Review, or underline the special aspects that have been addressed through the manuscript. In the actual shape, I cannot see the relevance of your Review from this aim of the study
L71-80 Information provided (inclusion/exclusion criteria) here must be mentioned in the PRISMA flow chart, removing the duplicate information (i.e. L77, 79, etc.). This is exactly the purpose of the PRISMA diagram: to summarise the description of references selection in a single schematised figure. Detail better the PRISMA figure.
L102. It must be completed that the active compounds identified in the plants you have focused have specific actions, roles and proprieties, strongly correlated with their chemical structure, as it was well stated by Glevitzky I., et al. Statistical Analysis of the Relationship Between Antioxidant Activity and the Structure of Flavonoid Compounds. Rev. Chim. 2019, 70(9), 3103-3107. https://doi.org/10.37358/RC.19.9.7497
Furthermore, the content in active compounds are also strong related to the conditions for their cultivation/growing (climate, type/characteristics of the soil/soil management, harvesting time, etc). I suggest checking and referring to Samuel A.D. et al.., Enzymological and physicochemical evaluation of the effects of soil management practices, Rev. Chim. 2017, 68(10) 2243-2247. https://doi.org/10.37358/RC.17.10.5864 ; Bungau et al. Expatiating the impact of anthropogenic aspects and climatic factors on long term soil monitoring and management. Environ Sci. Pollut. Res. 2021, 202, 30528-30550. https://doi.org/10.1007/s11356-021-14127-7
Before Conclusion, I consider you must add a new section related exactly to the title: Comparison of the four plants, which is mandatory, as in the actual shape of the manuscript is no comparison, just data about each plant, separately. A summarising Figure related to the effects/actions of all 4 plants and a table making comparison between their main chemical active compounds would be very helpful, increasing the relevance of the text.
Author Response

(The authors gave the same response as above.)

Round 2
Reviewer 3 Report
The authors responded to my requests.